# Investigating Simulated Cellular Interactions on Nanostructured Surfaces with Antibacterial Properties: Insights from Force Curve Simulations

**DOI:** 10.3390/nano15060462

**Published:** 2025-03-19

**Authors:** Jonathan Wood, Dennis Palms, Quan Trong Luu, Krasimir Vasilev, Richard Bright

**Affiliations:** 1Academic Unit of STEM, University of South Australia, Adelaide, SA 5095, Australia; jonathan.wood@unisa.edu.au; 2College of Medicine and Public Health, Flinders University, Adelaide, SA 5042, Australia; dennis.palms@flinders.edu.au (D.P.); luu0044@flinders.edu.au (Q.T.L.); krasimir.vasilev@flinders.edu.au (K.V.)

**Keywords:** atomic force microscopy, force curve, nanomechanical, van der Waals, nanostructure, antibacterial

## Abstract

This study investigates the simulation of interactions between cells and antibacterial nanostructured surfaces. Understanding the physical interaction forces between cells and nanostructured surfaces is crucial for developing antibacterial materials, yet existing physical models are limited. Force simulation studies can simplify analysis by focusing on mechanical interactions while disregarding factors such as bacterial deformation and complex biochemical signals. To simulate these interactions, Atomic Force Microscopy (AFM) was employed to generate force curves, allowing precise monitoring of the interaction between a 5 µm spherical cantilever tip and titanium alloy (Ti6Al4V) surfaces. AFM uniquely enables customized approaches and retraction cycles, providing detailed insights into attractive–repulsive forces across different surface morphologies. Two nanostructured surfaces, created via hydrothermal etching using KOH and NaOH, were compared to a Ti6Al4V control surface. Results demonstrated significant changes in nanomechanical properties due to surface chemistry and morphology. The Ti6Al4V control surface exhibited a 44 ± 5 N/m stiffness, which decreased to 20 ± 3 N/m on KOH-etched nanostructured (NS) surfaces and 29 ± 4 N/m on NaOH-etched NS surfaces. Additionally, surface energy decreased by magnitude on nanostructured surfaces compared to the control. The nature of interaction forces also varied: short-range forces were predominant on KOH-etched surfaces, while NaOH-etched surfaces exhibited stronger long-range forces. These findings provide valuable insights into how nanostructure patterning influences cell-like interactions, offering potential applications in antibacterial surface design. By tailoring nanomechanical properties through specific etching techniques, biomaterial performance can be optimized for clinical applications, enhancing antibacterial efficacy and reducing microbial adhesion.

## 1. Introduction

A high rate of surgical site infections occurs in knee arthroplasties, at over 4.5%, and around 2% for total hip arthroplasties. These are the recorded data from 6111 total joint arthroplasties performed over 2 years in the US [1]. In the UK, infections related to knee arthroplasty impact about 4% of patients and 15% of those undergoing revision knee replacements [2]. Infections are a serious problem caused by foreign materials placed inside the body. This includes implanted medical devices. These infections are often resilient against antibiotic treatments and the body’s immune system. Biofilm formation of bacteria protects these organisms. Once inside the body, the implant is met with a race between bacteria trying to establish a biofilm and the mammalian cells trying to integrate with the implant and kill off the bacteria. Host cells win this race if they successfully establish themselves on the implant and occupy its surface, leading to tissue integration around the implant, and minimizing the risk of bacterial biofilms forming. A permanent method to stop bacterial infection of an implant uses chemically grown nanoscale structures to kill bacteria on contact by mechanical interaction [3].

There is a lack of information on the short- and long-range forces between cells and antibacterial surfaces. These cells include bacterial and the body’s (mammalian) cells. Knowledge of these forces is necessary to understand the influence and approach conditions of cells from a distance to initial adhesion. Attractive long-range forces, such as Van der Waals forces (VDWs), significantly influence drawing cells toward attachment. These forces, named after the Dutch scientist Johannes Diderik van der Waals, are weak forces that act over a distance and are responsible for the attraction between molecules [4]. Longer-range forces are weaker than shorter-range forces, such as those measured in AFM force curves. These graphs show the force between the AFM cantilever tip and the surface as a function of the tip’s position [5]. However, as the name suggests, there is an attractive influence at greater distances from the sample substrate.

The interplay between short- and long-range forces significantly impacts the strength and dynamics of reversible cellular contacts [6,7]. Attachment and adhesion of bacterial and mammalian cells have been addressed [8]. However, the passage of cells in the long-to-short range has not. AFM force curves can provide several nanomechanical values between the cantilever and substrate surface. In particular, force curves can measure adhesion interactions between the AFM cantilever tip and surface or surface-bound features, including short- and long-range forces, capillary forces, and surface energy [9,10]. Information that can be gained from these measurements is the intensity, range, and distance of influence of these forces against an opposing force. Calculating adhesion measurements against a known cantilever spring constant equates to the overall adhesive forces of the surface. Short-range forces are more straightforward to quantify as the snap-to-contact behavior on approach and snap-off-contact on surface withdrawal are clearly defined and measured by a force curve [11]. Snap-to and snap-off behavior can be measured in AFM force curve mapping. Long-range forces, such as the VDW collection of forces, are not as straightforward to quantify [12,13]. VDW and similar forces influence force curve behavior over a graduating length, often beyond the range of the initial tip position, and can alter force intensity depending on the tip–sample separation [14].

The measurement of short- and long-range forces in force curves can become more intricate when interacting with non-uniform surfaces, introducing complexity to the analysis. Planar surfaces that feature crystalline faults, etching pits, or heavy scratches from polishing can change force curve behavior due to localized topographical and chemical variances in tip–sample contact and noncontact short-range forces [15,16]. A closely spaced nanostructured (NS) array on a substrate can significantly alter the behavior of force curves [17]. This is attributed to the variations in height, size, spacing, patterning, and deformation of these nanostructures [18]. As a result of these variations, the topmost layer of NS, which primarily interacts with an AFM tip, can be considered a pseudo-surface and may exhibit a preference for either short-range or long-range forces. Understanding the force behavior of nanostructured surfaces is crucial for developing antibacterial strategies, as it offers valuable insights into the interaction and attraction forces involved in antibacterial NS surfaces. This knowledge can enhance our understanding of the mechanisms underlying the interactions between microorganisms and NS surfaces, thereby facilitating the development of effective antibacterial strategies and influencing the design of biomaterials [19,20].

This study used a spherical AFM tip to represent either a bacterium or mammalian cell interacting with the NS. A spherical AFM tip is significant, as it allows us to mimic the shape and size of many bacterial and mammalian cells, providing a more realistic model for our experiments. We first showed the relationship between short- and long-range forces between the spherical AFM tip against the as-received control (AR-Ti) and hydrothermally etched (HTE) NS titanium alloy surfaces. Primarily, how the contrast between NS surface area and morphology affects the forces toward an approaching cell is determined. A thorough understanding and precise control of the NS array enable analysis and customization to modulate surface attraction or repulsion. Short-range interaction, where the cell is in physical contact with the surface, does not account for its susceptibility to initially being attracted to the surface. This needs to be addressed in the overall attractive forces for a comprehensive understanding of NS–cell interaction compared to an AR-Ti surface of an antibacterial implant. Two etchant solutions, KOH and NaOH, created unique hydrothermally etched nanostructured surfaces on medical-grade 5 titanium alloy (Ti6Al4V). Water contact angle (WCA) and X-ray photoelectron spectroscopy (XPS) characterized these surfaces for wettability, followed by AFM force curve analysis. Tip contact and adhesion contact areas were calculated using Hertz, Derjaguin–Müller–Toporov, and Johnson–Kendall–Roberts models to signify initial contact with a non-uniform surface [3]. This research, which calculated the impact of three contact forces of 20, 50, and 100 nN by a 5 µm diameter spherical cantilever tip, has the potential to influence the development of antibacterial NS biomaterials significantly. The choice of these specific forces and the size of the AFM tip is based on the typical forces exerted by bacteria during adhesion and the size range of bacteria, allowing us to simulate realistic conditions in our experiments. Understanding the forces at play is crucial before clinical use, as it will determine the effectiveness, safety, and suitability of these materials for integration into medical devices and implants. This promising outlook underscores our work’s importance in advancing the antibacterial biomaterials field.

## 2. Materials and Methods

### 2.1. Materials

Sodium hydroxide, potassium hydroxide, and hydrogen chloride were purchased from Sigma-Aldrich, (St. Louis, MO, USA). Milli-Q water with a resistivity of 18.2 MΩ·cm was utilized as an ultrapure water source for all experiments.

### 2.2. Sample Fabrication

Ti6Al4V discs (10 mm in diameter and 3 mm in height) polished at a Ra of 0.5 µm were received from Hamagawa Industrial (M) SDN BHD, Kedah, Malaysia. To obtain surface NS, the HTE treatment for titanium alloy begins with preparing an etching solution consisting of a Milli-Q water–hydrogen chloride mixture with either a KOH or NaOH solute added. The hydrothermal method is performed in a Teflon-based container with the titanium alloy discs immersed in the etching solution, heated within a stainless-steel autoclave from 150 to 180 °C. The sample was cooled and immersed in Milli-Q water for 2 h to eliminate any remaining etching mixture. This meticulous step ensured the complete removal of residual traces of the etching solution from the samples, instilling confidence in the reliability of our research. Samples were stored in a plastic container to protect them from atmospheric impurities.

### 2.3. Characterization of HTE Surfaces by Water Contact Angle (WCA)

Surface wettability was assessed for AR-Ti, KOH NS, and NaOH NS using the sessile drop method with a contact angle goniometer (model RD-SDM02, RD Support, Edinburgh, UK). The contact angle was measured for Milli-Q water (5 µL) as the probe liquid and analyzed using the tangent fitting method with the Contact_Angle.jar plugin for ImageJ software (version 1.53f51, NIH, Bethesda, MD, USA).

### 2.4. X-Ray Photoelectron Spectroscopy (XPS)

The chemical composition of the top 10 nm of AR-Ti, KOH NS, and NaOH NS surfaces was characterized using XPS. Survey spectra were collected with a Kratos AXIS Ultra DLD spectrometer (Kratos Analytical Ltd., Manchester, UK), utilizing a magnetically confined charge compensation system and monochromatic AlKα radiation (hν = 1486.7 eV). The analysis area was 300 µm × 700 µm, with a pass energy of 160 eV, and data processing was conducted using CasaXPS software version 2.3.16 PR 1.6 (Casa Software Ltd., Teignmouth, UK). Binding energies were calibrated to the low-energy aliphatic C 1s peak at 285.0 eV.

### 2.5. Scanning Electron Microscopy (SEM) Analysis

The surface morphology of the KOH NS and NaOH NS was analyzed using SEM (Zeiss Merlin FEG-SEM, Zeiss, Jena, Germany) at 10 kV, 5.0 mm working distance, magnification from 5000× to 50,000×, and a stage tilted at 45°. SEM micrographs were imported into ImageJ 1.53 (NIH, Bethesda, MD, USA, accessed on 6 June 2023) to determine the dimensions. SEM was used to characterize the initial surface and structure for processing quality and confirmation. Micron- and nanometer-scale determination of specifically designed surface formations was verified before further surface analysis was sought, especially of the complex and detailed three-dimensional structures [21,22].

### 2.6. Atomic Force Microscopy (AFM) Measurements

All samples were rinsed in water and ethanol before AFM analysis. Analysis was performed in the air using a JPK NanoWizard III AFM with an attached Nikon Instruments Eclipse Ti series (Melville, NY, USA) inverted optical microscope in a JPK acoustic enclosure. Linux-based JPK software version 5 was used for scan data acquisition. Post-data analysis was performed using Gwyddion version 2.54 scanning probe microscopy freeware version 2.67, ImageJ version 1.54i, and Office Excel 365 ProPlus. Three types of diving board-style cantilevers were used: an NT-MDT NSG30 with a spring constant between 22 and 100 N/m; an NT-MDT NSG03 with a spring constant between 0.35 and 6.1 N/m; and a custom cantilever with a single crystal silicon (SiO_2_) spherical tip with a diameter of 5 µm, possessing a spring constant between 13 and 77 N/m. A 5 µm diameter tip was used as an intermediate size between bacteria and mammalian cells [23,24]. Bacteria generally range from hundreds of nanometers to a few microns [23], while mammalian cells, such as macrophages, average 20–30 microns [25].

### 2.7. Force Curves

The anatomy of a force curve is set up as a 2D plot, with the tip height concerning the sample on the *x*-axis and the cantilever deflection response on the *y*-axis. Attractive forces occur between the tip–sample at distances more significant than the repulsive interatomic and chemical forces. Over a separation of several nanometers, VDW forces dominate the tip–sample interaction [14]. Performing AFM analysis in air, an atmospheric water layer will be present with a thickness < 10 nm [26]. As the tip approaches the water layer’s surface, the capillary effect causes the water layer to wick onto the tip, forming a capillary bridge [27]. This effect creates a relatively strong force that overcomes the repulsive forces, causing a snap-to-contact action, pulling the cantilever into physical contact with the sample surface [9]. An opposing snap-off contact action generates short-range surface forces, including the atmospheric water layer, and prevents the tip from detaching from the surface until stronger forces act to release its contact. Data can be averaged across a sample by performing several force curves through the Force Mapping mode (Fmap). Force Mapping performs a pre-set number of evenly spaced force curves over a set surface scan area. Force curve data of the tip–substrate interaction can be derived from either fitting individual curves or batch processing over the entire scan area [28]. AFM-specific software packages, such as Gwyddion, can calculate values for elastic modulus, surface energy, adhesion, and stiffness across regions of varying composition [29].

### 2.8. Statistical Analysis

Analysis and data visualization were performed using Gwyddion software, version 2.54, for surface characterization and Microsoft Office Excel for statistical and graphical results representation. Each experiment was conducted in triplicate unless otherwise specified. Data are presented as mean ± standard deviation (SD) to express central tendency and variability, clearly representing the dataset and facilitating the comparison between experimental groups. Statistical analysis for WCA was performed using one-way ANOVA with Dunnett’s method for multiple comparisons. XPS data were analyzed using two-way ANOVA, with multiple comparisons corrected by the Tukey method in GraphPad Prism version 10.2.0 (GraphPad Software, Boston, MA, USA), accessed on 8 September 2024. A *p*-value of less than 0.05 was considered statistically significant.

## 3. Results and Discussion

### 3.1. SEM

Figure 1 compares the complex NS surfaces imaged by SEM and AFM. The tilt of the sample in SEM changes the perspective compared to the *z*-axis mapping by the cantilever tip in AFM. The resolution achieved by AFM using a nanometer-sized tip cannot match the level of detail provided by SEM, which utilizes electrons for imaging. Figure 1a,b compare the KOH NS surface between SEM and AFM scans. The clumping behavior of groups of collapsed NS can easily be observed in the SEM scan (Figure 1a). The corresponding AFM image, Figure 1b, shows the upper structure of the NS’s non-uniform array over 5 µm × 5 µm. Figure 1c,d show the SEM to AFM contrast of the NaOH NS surface. The single NS morphology and angle can be seen in the SEM image in Figure 1c and can be compared to the very different KOH NS in Figure 1a. The smaller diameter, higher density, and point-like morphology are observed in the AFM scan (Figure 1d). The KOH NS surface appears vastly different between the SEM and AFM scans and different again when changing the AFM cantilever tip from a 20 nm diameter conical tip to a 5 µm diameter sphere. The shape of the tip mirrors the NS surface [30]. The 5 µm diameter SiO_2_ spherical-tipped cantilever can simulate the movement of a microsphere or microparticle, such as a cell, over the NS topmost layer. Morphology of the KOH-etched NS was measured from SEM images at 340 ± 175 nm in height, 83 ± 32 nm in diameter at mid-height, and an average spacing of 544 ± 150 nm. Conversely, NaOH NS measured 367 ± 80 in height, a diameter at mid-height of 62 ± 23 nm and an average spacing of 182 ± 48 nm [31]. Although the SEM images capture high-resolution details of the nanostructured surface, AFM allows for physical interaction with the top layers of the nanostructures, yielding various nanomechanical values.

### 3.2. Surface Characterization

WCA was measured to assess the hydrophilicity of samples before and after hydrothermal etching. The NaOH NS surface (7.6 ± 1.2°) and KOH NS surface (9.7 ± 1.1°) had significantly lower WCA compared to AR-Ti (76.0 ± 5.0°) and *p* < 0.0001, indicating increased hydrophilicity (Figure 2a). There was no significant difference between the NaOH NS and KOH NS surfaces (*p* > 0.05). XPS analysis (Figure 2b) showed significantly higher oxygen levels on KOH NS (49.9 ± 4.8%) and NaOH NS (50.3 ± 2.2%) surfaces compared to AR-Ti (36.8 ± 2.0%), suggesting the formation of a thicker TiO2 layer (*p* < 0.0001). This also reduced vanadium and aluminum concentrations, which may help mitigate concerns about their toxicity in implantable biomaterials. Additionally, small amounts of potassium and sodium were incorporated from the alkaline treatments.

Figure 3a displays a representation of the AFM force curve plot. The AFM software (Gwyddion software version 2.54) maps the cantilever’s movement and flexion over the force curve process, allowing data to be calculated at different points of the force curve [32]. The plot in Figure 3a displays the progression of the force curve. The cantilever tip begins its approach toward the surface at point A. After making contact at point B, the tip presses into the surface until the pre-set force value is achieved. The tip reduces the force on the sample surface as it moves upward along the *z*-axis. During retraction, C remains in contact with the surface until the adhesion breaks, and D returns to its original position when the tip withdraws from the surface. Figure 3b relates the labeled plot actions mentioned to the actions of the cantilever over the process.

### 3.3. Roughness Analysis

In contrast to the relatively low RMS roughness of the silicon wafer and glass slide at 100 pm and 200 pm, respectively, a higher roughness of the polished AR-Ti sample at 4–5 nm was primarily due to the high number of scratches, shown in Figure 4, of AFM tapping-mode topography scans at 20 µm × 20 µm, 5 µm × 5 µm, and 1 µm × 1 µm. The 20 µm × 20 µm AFM topography scan in Figure 4a shows the significant scratches from polishing. The larger scratches are several microns wide and were measured by the AFM cantilever tip to be up to 75 nm deep, measured in plot Figure 4b. Figure 4c displays magnified scratching from polishing over a 5 µm × 5 µm scan area. The plot in Figure 4d displays the more significant scratches, measured in microns, and highlights numerous more minor scratches. Figure 4e and accompanying plot Figure 4f at 1 µm × 1 µm magnify the morphology of the scratched surface with widths in the hundreds of nanometers, with spacings averaging 200 nm. Surface roughness at this scale can directly affect the morphology of the less than 100 nm diameter NS present on the surface. Moreover, the large-scale roughness of the AR-Ti sample heavily influenced the tip–sample contact area, affecting the measured surface energy at 70–100 mJ/m^2^ compared to 30–60 mJ/m^2^ for the silicon control. Surfaces with a higher surface energy allow for easier cell attachment [34]. Roughness was calculated using Gwyddion software version 2.54, focusing on one dimension, the *z*-axis. The roughness average was calculated as the average deviation of all points from the selected mean line over the evaluation length. The root-mean-square average was calculated from the measured height deviations from the mean line within the evaluation length [35].

### 3.4. Adhesion Force

The adhesion force required to “pull off” the spherical cantilever tip from a substrate can be calculated using two main contact mechanics theories: the Johnson, Kendall, and Roberts model (JKR); and the Derjaguin–Müller–Toporov model (DMT) [36,37]. This paper will consider only the DMT theory, as this model fits the experimental parameters. Tabor found that the JKR and DMT theories apply to opposite extremes of the relationship between surface force and sample compliance [38]. The JKR model applies to systems with large particles, high surface energies, and compliant materials. The original paper provides examples of particles with diameters measured in centimeters [39]. The DMT model was applied to small particles with high stiffness and a small radius of curvature. It also considers the interaction forces outside the contact area, as molecular forces act in a ring of noncontact adhesion. In the original paper, particle radius was given in µm [40]. The relationship between these two models is shown in Equation (1), where n = 1.5 for the JKR model and n = 2 for the DMT model [3,41], where n is a predetermined constant depending on the selected mode [37].(1)$$F_{ADH}=n\pi a_{0}W_{ADH}$$

At a zero load, that is, when F = 0 at the cantilever tip pull-off, Equation (2) gives the contact radius, $a_{0}$.(2)$$a_{0}={\left(\frac{12\pi R^{2}\gamma }{K}\right)}^{1/3}$$

Surface energy values (γ) were calculated from the work of adhesion values on the pull-off stage of the force curve operation [42], which was measured at 0.018 N/m for silicon. Young’s modulus of silicon was referenced at 169 GPa [43]. AR-Ti surface energy was measured at 0.103 N/m, and Young’s modulus is referenced at ~110 GPa [44]. The force of adhesion values in Table 1 was calculated using Equation (1) with a spherical tip-to-substrate contact radius calculated from Equation (2). Young’s modulus values (K) were required in Equation (2). Ti6Al4V NS Young’s modulus values have not been knowingly referenced and could not be calculated from acquired force curve values. An approximate value was calculated using the contact radius at zero load, Equation (2), and calculating the percentage of coverage of the KOH and NaOH NS surface. Values for *a*_0_ were 100.4 ± 0.003 nm for silicon, 882.5 ± 0.06 nm for the AR-Ti surface, 446.9 ± 0.008 nm for KOH NS, and 381.0 ± 0.004 nm for NaOH NS, all shown in Table 1. The DMT model parameters fit the experimental parameters as defined above. The JKR model was included in Table 1, as this may relate to an interaction between larger cells on the nanostructured surfaces. The DMT calculations resulted in a higher force of adhesion value between the tip–sample due to a higher n-value, as shown in Equation (1). As expected, KOH NS and NaOH NS force-of-adhesion values were lower than the AR-Ti sample (*p* < 0.001). A heavily reduced contact area between the spherical tip and the substrate was the primary reason for this, along with the change in wettability and surface energy [45,46]. Force curve adhesion is related to the “snap-off-contact”, where the cantilever tip separates from the surface [47]. This section of the force curve process is displayed in Figure 3, part C. Raw values of the snap-off-contact (SOC) force and distance, measured using Gwyddion version 2.54 and ImageJ software, version 1.54i, were obtained from individual force curves. Table 2 displays these values.

### 3.5. Tip Contact Area

Calculations of the contact region of the 5 µm diameter SiO_2_ spherical cantilever tip in relation to the surface scratches or the contact area based on the cantilever’s applied force by the surface energy, sphere radius, and spring constant are shown in Figure 5. This utilizes the Hertz model that negates adhesion effects and uses a zero applied force parameter. Displacement at zero force infers no deformation of the interacting sphere, as contrasted to cell deformation under an external load [48]. A simplified correlation between the silicon spherical tip and a spherical cell is the initial contact, at zero load, with no deformation, and how the contact area may be affected by the surface morphology and surface interaction. This has been previously simulated [48]. The area of adhesion of the outer blue circle in Figure 5 was calculated from Equation (1), with n = 1.5, in the Johnson, Kendall, and Roberts model, which relates to larger-scale soft surfaces, such as large cells or biological tissues. The adhesion work term *W*12 can be approximated as 2√*γ*1*γ*2, basing the interaction energy on the tip–sample system’s surface interaction energy [42]. The DMT model, shown by Equation (1), with n = 2, accounts for interaction forces outside the immediate contact area, such as capillary adhesion, *π*(*W*)*R*, and relates to stiff and small contact areas [49].(3)$$F_{adh}=2\pi RW$$

Figure 5d–i display theoretical contact areas of a 5 µm diameter spherical cantilever tip at an applied force over a polished Ti6Al4V sample surface. Figure 5d calculates the contact area at a 100 nN applied force, with Figure 5e magnifying this region. The inner circle represents the adhesive contact area, calculated from Equation (3). The outer blue circle is the overall contact area, calculated by the Derjaguin, Muller, Toporov, and Maugis theory (DMT-M), Equation (4). Figure 5f,g calculate the contact and adhesive contact area at 50 nN and Figure 5h,i at 20 nN. Equation (3) can be modified to remove the dependence of the *W* effects to simplify and relate the contact between the substrate surface and the spherical tip without secondary adhesive bridge interactions and remove interaction forces outside the strict sphere–planar contact. The DMT contact area was compared to the JKR contact area in Figure 5j,k. This modified Equation is displayed as Equation (1). Equation (1), measuring at F = 0 contact, calculates the tip DMT contact area as 96 nm^2^ compared to the area calculated by JKR as 988, 622, and 378 nm^2^ for applied forces of 100, 50, and 20 nN, respectively. At low loads, with *F* close to ±*F*_adh_, as *F* approaches the opposing pull-off force, the displacement, *δ*, changes from positive (compression) to negative (tension). Zero displacement (*δ* = 0) occurs not when *F* = 0 or *a* = 0 but when *F* = 0.89*F*_adh_ (negative value) and *a* = 0.76*a*_0_ [47].(4)$$A_{DMT-M}=\pi {\left(\frac{2\pi W}{K}\right)}^{2/3}R^{4/3}$$
where $K=\frac{4}{3}{\left[\frac{\left(1-v_{1}^{2}\right)}{E_{1}}+\frac{\left(1-v_{2}^{2}\right)}{E_{2}}\right]}^{-1}$, *R* is the radius of a spherical NP, and *W* is the work of adhesion.

Due to the AR-Ti surface being scratched due to polishing, it was expected that these scratches would vary force curve measurement values of adhesion, *W*. Surface energies dependent on the spherical tip contact area over these scratch regions will not only change the overall area of contact but also vary the evenness of wetting and short-range surface energy. This is displayed in the schematics in Figure 5, where the red section of the spherical tip represents the contact area to the surface. Figure 5a displays the maximum contact area between an ideal spherical tip and a flat, featureless surface with no scratches. Figure 5b shows the reduction in tip contact area due to a scratch channel compared to Figure 5a. This reduction in contact was highly dependent on the width, depth, angle, shape, number of scratches, and region of contact with the spherical tip. Any number of these parameters can alter the contact mechanics and vary the force curve measurement values. Capillary effects are also affected between the tip and substrate, shown in Figure 5c, and can change the force curve snap-off-contact force and distance values, as well as the snap-to-contact distance and force [36].

Upon retraction of the cantilever from the surface, the snap-off-contact force was consistently larger in magnitude than the snap-to-contact due to the formation of chemical bonds during the contact time, which was the deformation stage of the force curve. Meniscus forces and larger contact areas due to elastic and plastic deformation of the surface are other factors that increase the magnitude of snap-off-contact behavior. These can vary significantly over damage, faults, and chemical changes on the substrate surface or the cantilever tip. On the AR-Ti surface, polish scratches can be seen to heavily affect the snap-off contact and work of adhesion results in Figure 6a,c. The work of adhesion measures the snap-off-contact area in the force curve, Equation (5), which is the energy required to adhere the cantilever to the surface until opposing forces are higher than *Fadh*.(5)$$W=F_{ADH}^{2}/2k_{C}$$

The adhesion force combines the electrostatic force, VDW, meniscus, capillary forces, chemical bonds, and acid–base interactions. The VDW is a combination of the Keesom potential between two permanent dipoles, the Debye potential (Maxwell equations relating electromagnetic vectors), and the London potential (electron fluctuation influencing other electrons forming instantaneous dipoles, with the magnitude described by Haymaker constant) [36]. The meniscus force is the formation of a liquid (water) neck between the tip–sample from atmospheric water on surfaces with adhesion forces commonly in the 10–100 nN range. Meniscus forces are difficult to experimentally determine against surface roughness, tip–sample contact area, deformation effects, and the contribution of the other attractive force [5,36,50]. These combined factors make it difficult to separate the sole cause of Figure 6a–c’s variation in *W*. However, significant changes in surface contact between the tip and substrate due to surface damage are a primary factor, as simulated and calculated in Figure 5. Three *W* force curve areas are shown in Figure 6 over three separate contact regions of an AR-Ti surface. With a large area of adhesion as a measure of the *W*, it measures a sizeable pull-off force at 1.2 µN over a distance of 28 nm (Figure 6a). This was due to the contact area of the tip being on a part of the surface region with a minimal number of deep scratches. Figure 6b,c have reduced contact areas from an increase in large and deep scratches that measure a reduced adhesion force from a drop in consistent surface contact area, which relates to a drop in capillary adhesion effects, as shown in Figure 5’s schematic. Reduced snap-off-contact forces for Figure 6b,c are 0.46 µN over 12.3 nm and 0.25 µN over 9.5 nm, respectively. Depending on the contact and adhesive contact of the spherical tip, a wide range of nanomechanical measurements can be compared to the increased roughness of the sample surface.

The DMT contact area measured on KOH NS for regular cantilever forces of 100 nN was 106.8 nm^2^, 133.2 nm^2^ for 50 nN, and 132.9 nm^2^ for 20 nN. These contact area values do account for the non-uniform contact due to NS height differences, gaps between the NS, and additional contact from higher NS. Figure 7a shows the region of contact with a blue circle at 100 nN applied force with an area of 106.8 nm^2^. The positioning of this circle, as indicated by the magnified image in Figure 7b, sits halfway off an NS. Careful analysis of the surrounding surface structures shows that the tip landing on a region of uniform contact was unlikely. Figure 7c–e show a small range of non-uniform force curve contact with the KOH NS surface, affecting both approach and withdrawal from the surface at short distances. The non-uniform initial contact and withdrawal behavior was due to several factors [51], including the movement of the nanostructures, slippage of the tip, contact with several NSs, several adhesive contacts, and long and short-range force interactions [51].

Based on the above methodology, we also measured the tip contact area on NaOH NS using an NT-MDT NSG03 cantilever with a conical tip of the manufacturer’s quoted radius of <10 nm. As for the KOH NS scan in Figure 7, the region of DMT model contact was for a region of contact at an applied tip force of 100 nN. Individual NaOH NSs have a smaller diameter and spacing between the structures, as seen in Figure 8a. The DMT contact area using the NSG03 conical cantilever tip was the same as for the KOH NS, which was 106.8 nm^2^ at an applied force of 100 nN, represented by the blue circle in Figure 8a,b. Figure 8b, a magnification of the image in Figure 8a, shows that the area of contact encompasses more than one NS. This indicates that a spherical cell close to the diameter of the silicon cantilever tip will be more stable on this nanostructured surface than the KOH surface. The more contact points per unit contact area relate to a reduction in force per NS.

### 3.6. Comparison of KOH NS to NaOH NS

Titanium alloy KOH and NaOH NS possess a different topography from each other in patterning and individual NS areas, as recognizable in Figure 9a,b’s height scans. Ti6Al4V NaOH NS coverage was 43.2% of the 5 µm × 5 µm scan area (Figure 9c) and 50.6% NS coverage for KOH NS of the same scan size (Figure 9d), measured with a <10 nm radius conical cantilever tip. From the AFM images, the individual NS for KOH appears visually more prominent than the NS for NaOH. This was primarily due to the fusion of the KOH NS during the HTE process. Both NS diameters have been measured at 83 ± 32 nm for KOH and 83 ± 30 nm for NaOH [31]. ImageJ software version 1.54i was used to create binary images from the NaOH and the KOH nanostructured images from Figure 9a,b. The binary images are shown in Figure 9c,d, with the raised nanostructure features designated as white and the background designated as black. Some threshold values may filter or exclude lower NS features [52]. NaOH NS, sometimes referred to as nanograss or nanospikes [53,54], has a higher density (32 ± 9 per µm^2^) compared to KOH (8 ± 2 per µm^2^) [31]. KOH NSs cover much larger areas, as they are often clumped together at an angle between 45 and 90° to the surface plane, creating several hundred nanometers of vast structures. KOH NS also features higher NS “antennas”, which rise above the clumping region. The 2D plots in Figure 9g show that the NaOH NS spacing was tight compared to KOH NS (Figure 9f), with the SEM average spacing measured at 182 ± 48 nm and 544 ± 150 nm, respectively [33]. Analysis of a single image measures tip penetration for the KOH NS, as seen in Figure 9h, at a depth of nearly 700 nm and an NS spacing of 254 nm. The NaOH NS in Figure 9g shows a penetration depth of 204 nm and an NS spacing of 82 nm.

### 3.7. Stiffness Analysis

Stiffness values measured from the slope of the approach deformation stage of the force curve, part B of Figure 3a, were within a small error range for each AR-Ti sample. Glass, silicon, and titanium alloy surfaces possess high stiffness, with a force curve deformation slope in the mid to high 80° angle to the *y*-axis, as these materials had negligible deformation. Any deformation occurring from the force curve of these materials was more likely from overextension of the cantilever than from indentation or deformation of the substrate. Silicon and glass possess an elastic modulus of 169 and 72.5 GPa, respectively [43,55], while titanium alloy has a modulus of 110 GPa [56]. The stiffness values of the AR-Ti samples were compared to those of the KOH NS and NaOH, which had a modulus of approximately half that of the AR-Ti samples. As calculated from Table 3, KOH NS experiences a nearly 40% drop in stiffness compared to NaOH NS at a 50 nN set point. KOH NS was configured at a wide range of angles between 45° and 90° to the surface plane due to individual NS collapsing laterally during growth and usually clumping and fusing, as shown in Figure 1a. Due to this NS collapse, a wide aspect tip pushing on these bent structures sees greater flexion and collapse as the cantilever tip force was directed laterally as opposed to the NaOH NS that is orientated close to normal to the surface, simulated in Figure 10a, and KOH NS in Figure 10b, with an example of KOH NS collapse shown in Figure 10c. Given the expected higher deformation of cells compared to NS flexion, the contrast in stiffness between KOH and NaOH nanostructures may have minimal impact. The addition of other materials to the titanium alloy NS may offer additional improvements that alter the mechanical properties and reaction of the NS with an applied force [57,58]. These materials could be used to modify the stiffness, change the KOH NS collapsing and fusing behavior in manufacture, or change its elastic–plastic ratio [59]. An example includes alloying with transition metals such as nickel or niobium. Nickel–titanium alloys possess super-elastic properties [60,61].

### 3.8. Surface Energy Analysis

Surface energy values are high for glass due to its hydrophilic properties [26]. This was supported by the snap-to-contact forces, especially the snap-off-contact forces, being an order of magnitude greater than the hydrophobic silicon and AR-Ti surfaces, as shown in Table 2. Increased cantilever withdrawal velocities in the force curve procedure increase the surface energy between the tip and sample and decrease the snap-off-contact force and distance. The solid–vapor–liquid interfacial energy (sv/sl) from Equation (3) calculates *γ**s**v*/*s**l* for glass [39]. A low 0.5 µm/s withdrawal velocity measures a surface energy of 0.284 N/m. A 5 µm/s withdrawal velocity measures higher energy at 0.45 N/m. Contrast this at 1.57 × 10^−3^ to 1.146 × 10^−2^ N/m for the silicon control. Glass’s surface water layer creates increased capillary pull-off forces. A high cantilever withdrawal velocity creates a rapidly thinning capillary that breaks at the reduced tip–substrate distances. In contrast, at low velocities, the capillary bridge holds at greater distances with time before the capillary slowly thins out before breakage [26,62,63]. These controls can be compared to the *γ**s**v*/*s**l* values for the AR-Ti at 2.33 × 10^−2^ to 3.5 × 10^−2^ N/m for the KOH NS at 9.3 × 10^−4^ to 4 × 10^−3^ N/m, and the NaOH NS at 8.5 × 10^−4^ to 5.77 × 10^−3^ N/m. The lower capillary area contributed to lower surface energy for the NS, as shown in the schematic in Figure 5c. Over an approaching large particle, lower capillary-induced forces may aid in a reduction in adhesion [64].

Energy barriers at NS edges appear to have a negligible contribution to the overall surface energy affecting micron diameter structures from the reduced values for NS compared to the AR-Ti sample [65]. Higher surface energy is desirable for bone implants, as the increased wettability enhances implant surface–biological environment interaction. Cell spreading is enhanced on surfaces with higher surface energy [66,67,68]. Surface hydrophilicity is also a factor that determines biocompatibility and is dependent on surface energy. Increased hydrophilicity, such as the KOH and NaOH NS surfaces, improves bone formation [69]. This suggests that surface energy modulates the maturation of bone cells. Pure Ti surfaces form an oxide layer that makes the surface hydrophilic, binding atmospheric water and forming -OH and -O2- groups in the outer layer. Oxide surfaces can spontaneously nucleate calcium phosphate (apatite) layers upon contact with physiological fluids. Surfaces with high energy absorb inorganic ions or organic hydrocarbon atmospheric contaminates, resulting in altered surface composition and decreased hydrophilicity [69]. Snap-to-contact force and distances vary for all samples measured due to variations in surface crystalline orientation, roughness, and defects, with force reducing for rougher surfaces, as in the heavily scratched AR-Ti and the KOH NS. Snap-to-contact and snap-off-contact forces and distances measured an extensive range of values on each surface, with lower values for NS due to reduced contact area from a patchy surface underneath the tip.

### 3.9. Short- and Long-Range Forces on NS

The cantilever tip was subjected to three main force types: the cantilever spring force; the short-range contact forces from the sample surface; and the long-range forces, such as the VDW and electrostatic. Without short-range repulsive forces, the tip positioning at equilibrium will balance VDW, electrostatic, and cantilever spring forces [70]. Measuring force curve data for NaOH NS, where long-range forces dominate over shorter-range forces, was complex, as the noncontact approach and withdraw region are not linear, as shown in Figure 11. Due to the noncontact approach line being under the influence of the longer-range forces, there was no measurable snap to contact. The approach deformation line will also be subject to curvature rather than the straight deformation line, as there was still a transition or combination of long- and short-range forces. Withdrawal measurements of the adhesion force and distance, and the *W* were measured from the initial contact point of the lowest approach point (Figure 11). This point was taken as the tip–sample contact, and there was no clear snap to contact, as observed for the KOH NS force curves. Above this point was the noncontact region between the cantilever tip and the sample. Due to the NS geometry and reduced surface coverage, longer-range forces dominate from the multi-distance contribution between the topmost layer of the NS and the greater exposed base surface from reduced NS coverage. Challenges arise when calculating long-range forces, as multiple papers elucidate the various factors contributing to these forces, including capacitive, electrostatic, or VDW effects. Long-range surfaces consist of dispersion forces, London forces, electrodynamic forces, and dipole forces [36,71]. Grouping these as dispersion forces, they consist of a third of the important contributions of the VDW. These are effective at surface-to-surface distances above 10 nm at interatomic level spacings and 0.2 nm. Dispersion forces are not additive. The force between two surfaces is affected by other nearby surfaces, such as other NS, contributing to the total interaction energy [72]. These can include localized capillary forces and combinations of electrostatic and VDW from the surrounding surface morphology [73]. VDW decays at 1/*r*^6^ and 1/*r*^7^ toward a 100 nm tip–sample separation [74]. Figure 11 displays attractive decay past 800 nm separation, indicating a combination of long-range forces other than VDW, such as electrostatic or capacitive forces [13,75,76,77,78,79].

Relating to weaker, albeit longer-range, forces of the NaOH NS [80] reduces the favorability of this surface as an antibacterial interface, as cells will be exposed to attractive forces at greater distances. In determining shorter-range adhesion in snap-to-contact separations, it was difficult to accurately compare the two types of NS due to the lack of a clear snap-to-contact starting point for the NaOH force curve in Figure 11. Lower surface energy for the NaOH NS, with the middle value of the range being 10.2 pN/nm compared to 14 pN/nm for KOH, would suggest lower cell adhesion. Further exploration is needed to examine the strength of cellular contact by comparing the snap-to-contact distance and adhesion energy of the two NS surfaces. This analysis would offer valuable insights into the topic.

**Figure 11 nanomaterials-15-00462-f011:**
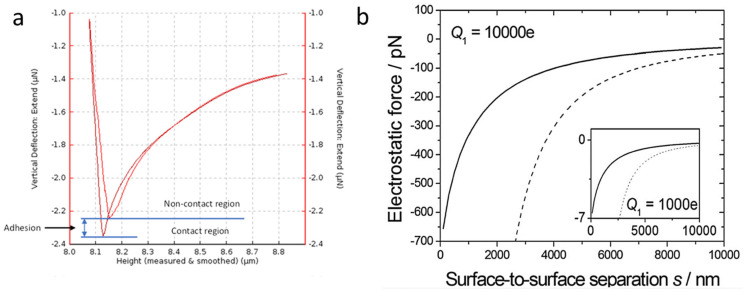
(**a**) Ti6Al4V NaOH NS force curve defining *W* area. The difference between the contact and noncontact regions is highlighted, measuring approximately 100 nN of force on the *y*-axis and a height difference of approximately 40 nm on the *x*-axis. (**b**) A plot displays the relation between short- and long-range (chemical) forces (VDW) [81].

Measuring surfaces with a micron-scale spherical cantilever tip, such as a 5 µm SiO_2_ tip, effectively mimics cellular interactions with complex nanostructures (NSs). Surface roughness and contact levels significantly influence adhesion forces, surface energy, and capillary effects, allowing for calculations of initial contact areas and adhesion zones based on known densities. Morphological changes in antibacterial NS surfaces, including alterations in structure size and decreased localized coverage from approximately 51% to 43%, transition the interactions from short-range to long-range. NaOH NS surfaces attract cells over hundreds of nanometers through long-range VDW and electrostatic forces, while KOH NS surfaces exhibit predominantly short-range forces. KOH and NaOH NS have lower surface energy and stiffness than AR-Ti surfaces, which have a surface energy of 61.5 pN/nm, compared to 14 pN/nm for KOH NS and 10.2 pN/nm for NaOH NS. KOH NS’s higher cellular attraction and lower stiffness enhance its suitability for antibacterial implants. Stiffness decreases from 44 N/m for the AR-Ti to 20 N/m for KOH and 29 N/m for NaOH NS, with corresponding reductions in surface energy from 10⁻^2^ to 10⁻^3^ N/m. These findings show that surface morphology modifications effectively regulate adhesive forces, stiffness, and interactions. While the JKR and DMT models provide valuable insights into adhesion forces, their applicability to nanostructured titanium surfaces has limitations, as the DMT model assumes negligible deformation, and the JKR model is typically suited for soft materials; additionally, surface roughness may introduce deviations from ideal theoretical assumptions, which could be further refined through experimental validation or alternative modeling approaches.

Future research could focus on optimizing the fabrication processes of NS to enhance their antibacterial properties further and exploring the integration of these surfaces into various biomedical devices. Additionally, investigating the long-term interactions between nanostructured surfaces and biological systems will provide deeper insights into their performance in vivo. Understanding these dynamics will facilitate the development of next-generation implants that promote better integration with host tissues, reduce infection rates, and improve patient outcomes in regenerative medicine.

## 4. Conclusions

In conclusion, measuring surfaces with a micron-scale spherical cantilever tip, such as the 5 µm SiO_2_ tip, provides valuable insights into the interaction dynamics of cellular and nanoparticle systems. By eliminating the complexities of cellular deformations and keeping the interaction at a low complexity, this approach allows for a controlled investigation of how surface roughness and contact levels influence adhesion forces, surface energy, and capillary effects. The findings highlight that changes in NS morphology and surface coverage significantly impact the force interactions, shifting dominance between short- and long-range forces. The KOH NS surface displayed force curve behavior related to the control surface. Forces such as capillary effects and short-range forces dominated the control and KOH NS surfaces, while long-range forces appeared to dominate the NaOH NS surface. Pattern variances in these two NS surfaces will differ in their response to attracting local cells. The observed differences in surface energy and stiffness between KOH, NaOH, and AR-Ti surfaces reveal that specific nanostructured configurations, particularly KOH NS, offer advantages for antibacterial and other cellular applications by enhancing cell attraction and flexibility. These results suggest that customizing NS characteristics can enhance antibacterial surfaces for implant and biosensing applications, marking a significant advancement in developing next-generation biomedical devices.

The next step toward understanding cellular deformation behaviors and effects on nanostructured surfaces is to repeat experiments in liquid, such as a saline solution. The next step is to use 5 µm spheres of a lower elastic modulus material, such as polyurethane, that allows for spherical deformation. Each step will add a significant step in complexity toward a complete picture of cellular interaction with complex nanostructured surfaces.

## Figures and Tables

**Figure 1 nanomaterials-15-00462-f001:**
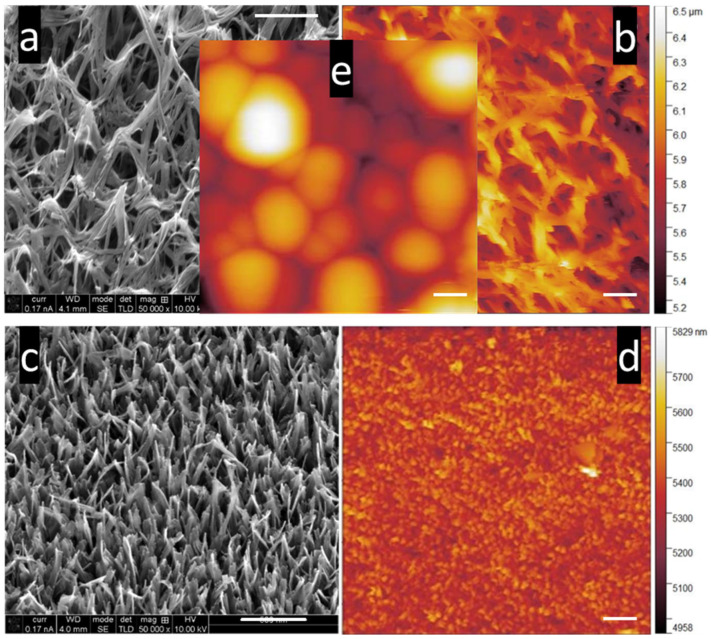
SEM image of (**a**) KOH NS and (**c**) NaOH NS; 5 µm × 5 µm AFM topography image of (**b**) KOH NS, (**d**) NaOH NS, and (**e**) KOH NS, scanned with a 5 µm diameter spherical-tipped SiO_2_ cantilever. The size of this scan area allows both the contrast of the surface and the morphology of the NS to be visualized clearly. Inset: 5 µm × 5 µm AFM topography image of KOH NS using a 5 µm diameter spherical cantilever tip. AFM images were processed and colored using Gwyddion software version 2.54. An NT-MDT NSG03 cantilever was used for AFM imaging of (**b**,**d**) with the following parameters: sensitivity, 29.3 nm/V; Q-factor, 154; amplitude, 8 × 10^−15^ m/√Hz; spring constant, 2.7 N/m; and set point 23 nN. A 5 µm diameter SiO_2_ spherical tipped cantilever was used in scan (**e**) with the following parameters: sensitivity, 25.8 nm/V; Q-factor, 292; amplitude, 8.6 × 10^−16^ m/√Hz; spring constant, 38 N/m; set point, 303.3 nN. The scale bar in the SEM micrographs and AFM images represents 500 µm.

**Figure 2 nanomaterials-15-00462-f002:**
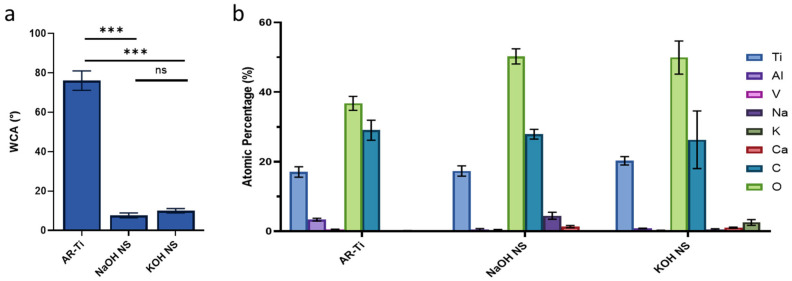
Surface characterization by (**a**) water contact angle, *** *p* < 0.001, ns = non-significant, and chemically by (**b**) XPS analysis. Data plotted as mean ± standard deviation and *n* = 3.

**Figure 3 nanomaterials-15-00462-f003:**
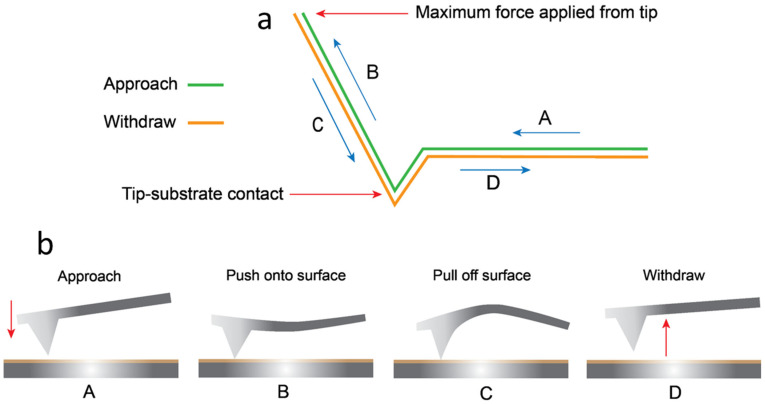
(**a**) Force curve 2D plot The blue arrows indicate the direction of the tip movement. (**b**) Primary steps of a force curve [33]. Red arrows show the direction of the force applied.

**Figure 4 nanomaterials-15-00462-f004:**
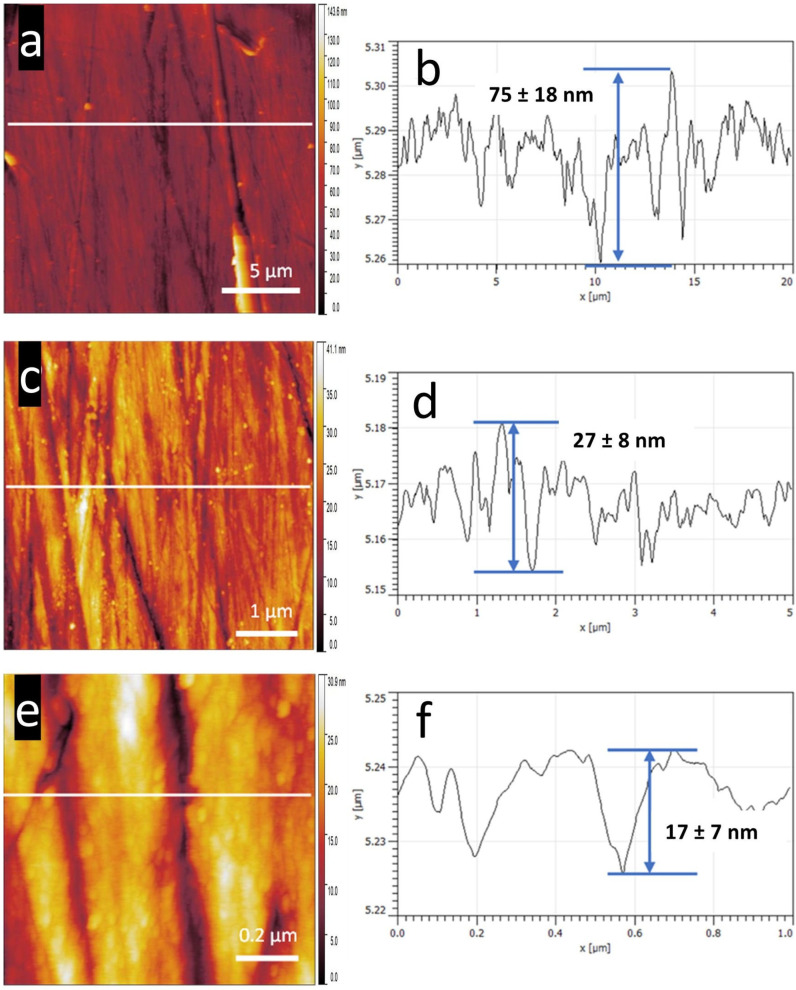
The AR-Ti height scan at 20 µm × 20 µm, and the (**b**) 2D plot across (**a**) taken at the horizontal white line shows a 75 nm change in *z*-axis topography with distances between significant scratches ranging from 400 to 1000 nm. Roughness was measured in Gwyddion version 2.54 at Ra 6.8 nm and RMS 10.6 nm. (**c**) AR-Ti scan at 5 µm × 5 µm, and (**d**) 2D plot across (**c**) at the horizontal white line showing a 27 nm change in *z*-axis topography with distances between more significant scratches ranging from 170 to 240 nm. Roughness was measured in Gwyddion version 2.54 at Ra 3.7 nm and RMS 5.1 nm. (**e**) AR-Ti scans at 1 µm × 1 µm, and (**f**) 2D plot across (**e**) taken at the horizontal white line, showing a 17 nm change in *z*-axis topography with distances between the more significant scratches ranging from 160 to 240 nm. Roughness was measured in Gwyddion version 2.54 at Ra 4 nm and RMS 5 nm. An NT-MDT NSG30 cantilever was used for AFM imaging with the following parameters: sensitivity, 38.6 nm/V; Q-factor, 513; amplitude, 6.8 × 10^−16^ m/√Hz; spring constant, 38.7 N/m; and set point, 32.2 nN. A higher spring constant cantilever was applied in this measurement to minimize atmospheric water layer effects and high response to surface cavities.

**Figure 5 nanomaterials-15-00462-f005:**
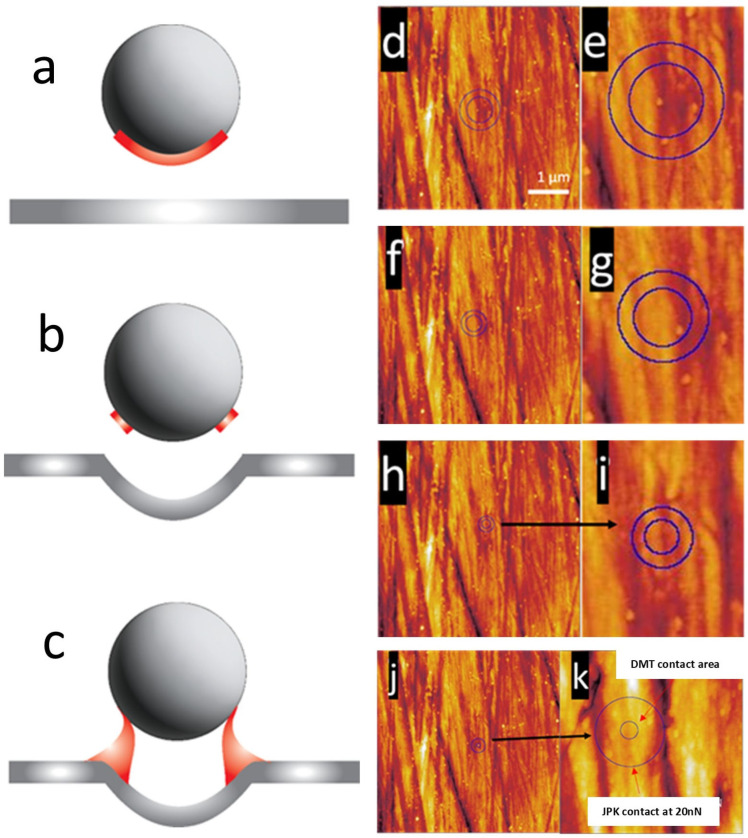
Schematic of spherical tip interaction with the AR-Ti surface: (**a**) contact on a flat, featureless region with the red area displaying the potential sphere–flat surface contact area; (**b**) over a scratch channel with the red area displaying the potential surface contact area; and (**c**) over a scratch channel showing a potential reduction in capillary effects. The 5 µm spherical tip contact area calculated through Equation (3). (**d**,**e**) Tip contact area of 987.5 nm^2^, and adhesion contact area of 622.1 nm^2^, as measured in ImageJ version 154i. with an applied normal force of 100 nN. (**f**,**g**) Tip contact area of 622.1 nm^2^, and adhesion contact area of 392 nm^2^, as measured in ImageJ version 1.54i with an applied normal force of 50 nN. (**h**,**i**) Tip contact area of 377.7 nm^2^, and adhesion the contact area of 212.8 nm^2^, as measured in ImageJ version 1.54i with an applied normal force of 20 nN. (**j**,**k**) DMT model calculated a contact area of 95.7 nm^2^ (inner blue circle) compared to the outer JKR model calculated contact area (outer blue circle) with the JKR model contact area of 377.7 nm^2^ for a contact force of 20 nN on (**j**) 5 µm × 5 µm scan and (**k**) 1 µm × 1 µm scale. The 5 µm diameter SiO_2_ spherical tipped cantilever used has the following parameters: sensitivity, 25.8 nm/V; Q-factor, 292; amplitude, 8.6 × 10^−16^ m/√Hz; spring constant, 38 N/m; and set point, 303.3 nN.

**Figure 6 nanomaterials-15-00462-f006:**
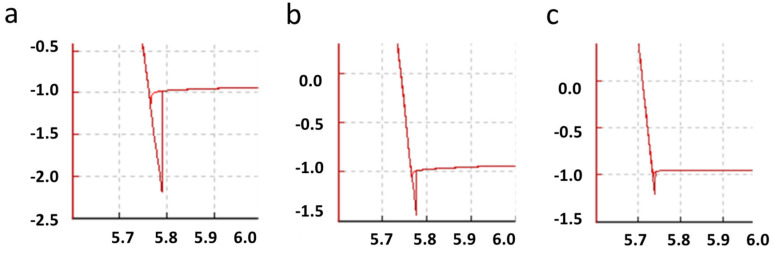
AR-Ti force curves with (**a**) a large adhesion contact area, (**b**) a medium adhesion contact area, and (**c**) a small adhesion contact area. The *y*-axis value is vertical deflection in µm, and *x*-axis is height in µm. The 5 µm diameter SiO_2_ spherical-tipped cantilever used has the following parameters: sensitivity, 25.8 nm/V; Q-factor, 292; amplitude, 8.6 × 10^−16^ m/√Hz; spring constant, 38 N/m; and set point, 303.3 nN.

**Figure 7 nanomaterials-15-00462-f007:**
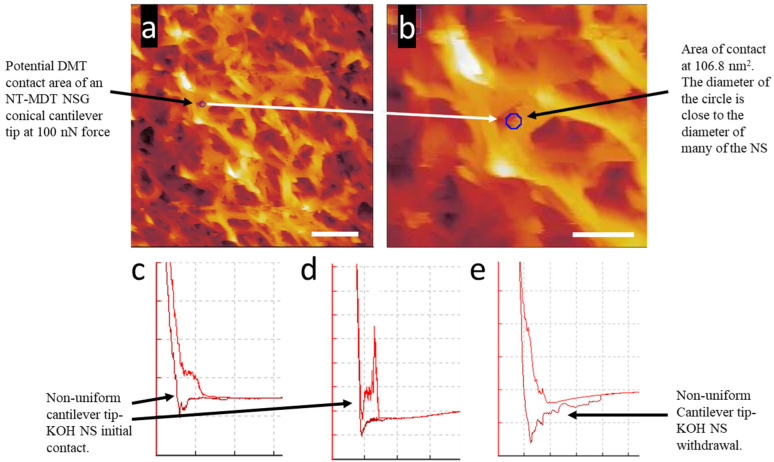
Contact for Derjaguin–Müller–Toporov on a KOH NS surface at an (**a**) 5 µm × 5 µm area (scale bar = 1000 µm) and a (**b**) 2 µm × 2 µm area (scale bar = 500 µm). The blue circle indicates the contact area from the conical-shaped cantilever tip at the maximum quoted diameter by the manufacturer of 20 nm. The blue circle, magnified in the smaller scan region (**b**), is the potential tip contact area on the KOH NS, which displays that the tip is likely to contact unstable and complex regions over the NS array. (**c**–**e**) Force curve 2D plots of non-uniform cantilever tip contact on a KOH NS surface. The plot’s *x*-axis is the cantilever tip movement in the *z*-axis, and the plot’s *y*-axis is the force experienced by the cantilever. The *y*-axis value is vertical deflection in µm, and *x*-axis is height in µm. An NT-MDT NSG03 cantilever was used for AFM imaging with the following parameters: sensitivity, 29.3 nm/V; Q-factor, 154; amplitude, 8 × 10^−15^ m/√Hz; spring constant 2.7 N/m; and set point, 23 nN.

**Figure 8 nanomaterials-15-00462-f008:**
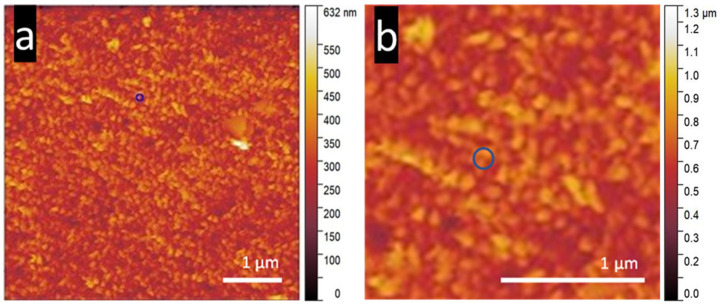
Contact for DMT on a NaOH NS surface at an (**a**) 5 µm × 5 µm area and a (**b**) 2 µm × 2 µm area. The blue circles indicate the contact area from the conical-shaped cantilever tip at the maximum quoted diameter by the manufacturer of 20 nm. An NT-MDT NSG03 cantilever was used for AFM imaging with the following parameters: sensitivity, 29.3 nm/V; Q-factor, 154; amplitude, 8 × 10^−15^ m/√Hz; spring constant, 2.7 N/m; and set point 23 nN.

**Figure 9 nanomaterials-15-00462-f009:**
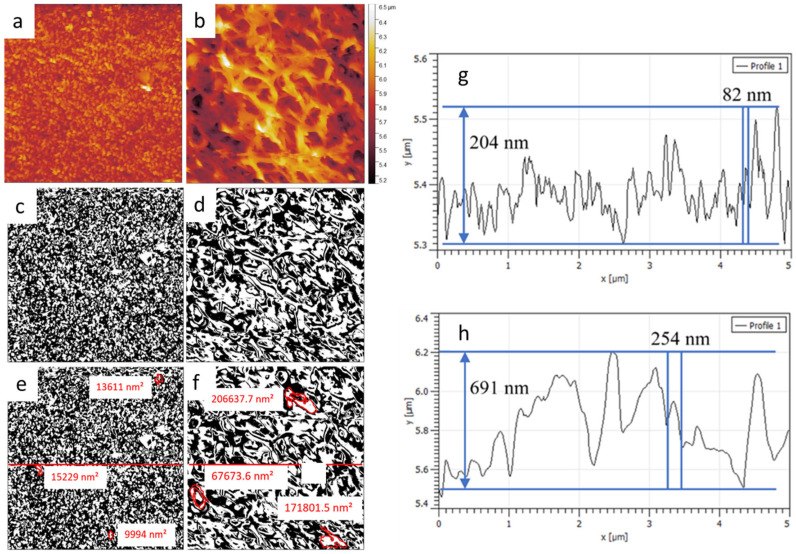
AFM topography scan of (**a**) NaOH NS and (**b**) KOH NS with scan sizes of 5 µm × 5 µm. Height image with a binary filter applied in ImageJ software version 1.54i, with white being the raised NS features measured by an NSG03 conical, 10 nm radius cantilever tip; (**c**) NaOH NS; (**d**) KOH NS; (**e**) measuring the area of three NaOH NS at 1.4 × 10^4^, 1.5 × 10^4^, and 1.0 × 10^5^ nm^2^; (**f**) measuring the area of three KOH NS at 2.1 × 10^5^, 6.8 × 10^4^, and 1.7 × 10^5^ nm^2^. Two-dimensional plots were measured across the red line in the binary images (**e**,**f**) for (**g**) NaOH NS and (**h**) KOH NS. A NT-MDT NSG03 cantilever was used for AFM imaging with the following parameters: sensitivity, 29.3 nm/V; Q-factor 154; amplitude, 8 × 10^−15^ m/√Hz; spring constant, 2.7 N/m; set point, 23 nN.

**Figure 10 nanomaterials-15-00462-f010:**
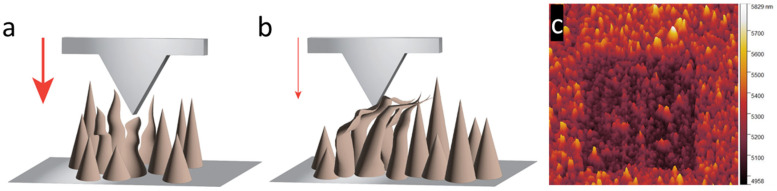
The deformation of the nanostructures is caused by the downward force exerted by the cantilever tip on (**a**) the NaOH NS and (**b**) the collapsed KOH NS. The red arrows highlight the downward force produced by the cantilever. The thick red arrow for the NaOH NS shows the higher force required for a buckling-type collapse, and the thinner arrow shows the less force needed to deform a long NS laterally. The deformation of the KOH NS can be seen in (**c**). A 1.5 µm × 1.5 µm area was scanned in contact mode with an NT-MDT NSG30 cantilever with a spring constant of 28 N/m. The damage can be visualized when a more prominent area tapping mode scan is performed.

**Table 1 nanomaterials-15-00462-t001:** Adhesion forces for JKR and DMT models were calculated from force curves at twelve random points on silicon, AR-Ti, KOH NS, and NaOH NS arrays. Results are presented as mean ± SD (*n* = 12, in N/m) to illustrate the wide variation between AR-TI and HTE surfaces.

Surface	JKR (N/m)	DMT (N/m)
Silicon	4.0 ± 0.6	5.3 ± 0.7
AR-Ti	185.3 ± 122.6	247.1 ± 163.5
KOH NS	13.4 ± 8.8	17.2 ± 11.8
NaOH NS	4.4 ± 3.5	5.9 ± 4.7

**Table 2 nanomaterials-15-00462-t002:** Comparison of force curve measurement nanomechanical values for three control and two Ti6Al4V NS surfaces. A custom cantilever derived these values with the following parameters: sensitivity, 30.9 nm/V; Q-factor 72; amplitude, 1.6 × 10^−14^ m/√Hz; spring constant, 1.5 N/m; and set point, 26 nN.

Samples	Surface Energy (pN/nm)	Snap-to-Contact Force (nN)	Snap-to-Contact Distance (nm)	Snap-off-Contact Force (nN)	Snap-off-Contact Distance (nm)
Si control	6.7–35	10–180	2–28	37–270	10–30
Glass control	850–1350	150–290	5–11	6700–10,600	149–245
AR-Ti (control)	6.9–130	86–190	9–15	550–830	17–20
Ti6Al4V KOH	4–24	1–20	208	22–96	15–22
Ti6Al4V NaOH	3.4–17	-	-	20–136	6–36

**Table 3 nanomaterials-15-00462-t003:** Stiffness values (N/nm) are derived from the rise–run of the approach deformation section of the relevant force curve. Force curve tip velocity throughout the force curve was performed from 2 to 0.25 µm/s. Measuring KOH and NaOH nanostructured surfaces with a force curve set point (maximum force of cantilever tip on the sample) of 100 and 50 nN. Values are average stiffness measured over 4–13 force curves, with standard deviation values in brackets. A NT-MDT NSG03 cantilever was used for AFM imaging with the following parameters: sensitivity, 29.3 nm/V; Q-factor 154; amplitude, 8 × 10^−15^ m/√Hz; spring constant, 2.7 N/m; and set point, 23 nN. Data presented as mean ± standard deviation.

Tip Velocity (µm/s)	NaOH 100 nN	NaOH 50 nN	KOH 100 nN	KOH 50 nN
2	27.3 ± 1.6	28.6 ± 1.6	24.1 ± 5.7	17.6 ± 2.1
1.5	30.7 ± 3.7	28.8 ± 1.4	23.5 ± 4.5	17.2 ± 1.6
1	31.3 ± 2.4	28.6 ± 1.6	25.2 ± 4.0	17.7 ± 2.1
0.5	30.0 ± 3.2	28.6 ± 1.6	27.4 ± 4.1	17.4 ± 2.8
0.25	30.9 ± 3.9	28.8 ± 1.0	31.6 ± 4.5	18.1 ± 2.5

## Data Availability

Data will be made available upon request.

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
