# Peer review of "Investigating Simulated Cellular Interactions on Nanostructured Surfaces with Antibacterial Properties: Insights from Force Curve Simulations"

_nanomaterials, 2025, doi:10.3390/nano15060462_

Round 1

Reviewer 1 Report

Comments and Suggestions for Authors

The article "Investigating Simulated Cellular Interactions on Nanostructured Surfaces with Antibacterial Properties: Insights from Force Curve Simulations" is devoted to the study of cell interactions with nanostructured antibacterial surfaces using atomic force microscopy (AFM). The article is well structured and contains a detailed description of the methods and results, but it has several significant drawbacks that may reduce its scientific value and clarity of data presentation (need major revision).
1. The paper analyzes mechanical interactions, but there is no experimental biological data (for example, how real bacteria or cells interact with these surfaces). The author uses a spherical AFM probe as a cell model, but does not explain how accurately it mimics the behavior of bacteria in real conditions.
2. The conclusion repeats part of the discussion, but does not offer clear conclusions and practical recommendations.
3. The article considers Van der Waals forces and electrostatic forces, but does not take into account capillary forces, which can significantly affect bacterial adhesion in a humid environment (which is critical for medical implants).
4. The article uses JKR and DMT models for calculating adhesion forces, but does not verify the applicability of these models to this type of surface. The DMT model assumes that the interaction occurs due to external adhesive forces, but does not take into account the deformation of nanostructures, which can lead to incorrect calculations. The JKR model, on the contrary, is more suitable for soft, highly deformable surfaces (for example, polymers), but the article does not show that the titanium surface behaves in this way.

Reviewer 2 Report

Comments and Suggestions for Authors

The manuscript titled “Investigating Simulated Cellular Interactions on Nanostructured Surfaces with Antibacterial Properties: Insights from Force Curve Simulations” by Wood, J.; et al. is a scientific work where the authors addressed the topology, adhesion and mechanical properties of films made of sodium hydroxide, potassium hydroxide and titanium alloys at the nanoscale. The most relevant outcomes found in this research could contribute to gain knowledge of the single-molecule interactions exerted by these films. The manuscript is generally well-written and this is a topic of growing interest.

However, it exists some points that need to be addressed (please, see them below detailed point-by-point) to improve the scientific quality of the submitted manuscript paper before this article will be consider for its publication in Nanomaterials.

1) Keywords. The authors should consider to add the term “antibacterial properties” in the keyword list.

2) Introduction. Could the authors provide quantitative data insights accordign the worldwide global burdens of infection diseases? This will significantly aid the potential readers to better understand the significance of this devoted research.

3) Materials & Methods. “2.6. Atomic Force Microscopy (AFM) Measurements” (lines 147-159). Did the authors characterize the tip radius prior the AFM data acquisition?

4) “2.7. Force curves” (lines 160-179). How did the authors distinguish between AFM tip-sample surface specific and unspecific events?

5) Why did the authors carry out the measurements in air conditions instead of liquid media mimicking the bacterial environment?

6) Results & Discussion. The adhesion force required to “pull off” the spherical cantilever from a substrate (…) This paper will only consider the DMT theory, as this model fits the experimental parameters” (lines 291-294). Why did the authors only consider the DMT model since it only takes into account nanoindenters with small tip radius whereas, in this work it was used a micrometer-size indenter?

7) Figure 7, panels a and b (line 456). The lateral scale bar should be added. Same comment for the Fig. 9 (line 507).

8) “3.7. Stiffness analysis” (lines 517-551). Here, I agree with the information provided by the authors albeit it needs to be discussed how the nanomechanics [1] of titanium alloy surfaces can be improved after the additon of certain metals [2]. This will serve to strengthen the message of this subsection.

[1] https://doi.org/10.3390/nano13060963

[2] https://doi.org/10.3390/ma17215161

9) Finally, where are the examination of cellular samples (living bacteria) to complement the current research?

10) “4. Conclusion” (lines 660-676). This section perfectly remarks the most relevant outcomes found by the authors in this work and also the promising future prospectives. It may be advisable to add a brief statement to remark the potential future action lines to pursue the topic covered in this research.

Round 2

Reviewer 1 Report

Comments and Suggestions for Authors

The manuscript can be accepted after minor changes.
Perhaps if a height scale is added to the AFM pictures, this will make the figures clearer.

Author Response

Reviewer 1

Perhaps if a height scale is added to the AFM pictures, this will make the figures clearer.

Thank you for your helpful comment on improving the clarity of our manuscript figures. We have added a height scale to Figures 1, 4, 8, 9, and 10.

Reviewer 2 Report

Comments and Suggestions for Authors

The authors did a great deal of effort to cover all the suggestions raised by the Reviewers. However, the point 8 from the previous report was not fully addressed and the suggested references should be included to highlight the information furnished in this statement.

Author Response

Reviewer 2

The authors did a great deal of effort to cover all the suggestions raised by the Reviewers. However, the point 8 from the previous report was not fully addressed and the suggested references should be included to highlight the information furnished in this statement.

Thank you for your valuable input. We acknowledged an oversight and have added the references as you suggested.

L554  “The addition of other materials to the titanium alloy NS may offer additional improvements that alter the mechanical properties and reaction of the NS with an applied force [60, 61].”